# Comparison of Bulk Polymeric Resin Composite and Hybrid Glass Ionomer Cement in Adhesive Class I Dental Restorations: A 3D Finite Element Analysis

**DOI:** 10.3390/polym16172525

**Published:** 2024-09-05

**Authors:** Alessandro E. di Lauro, Stefano Ciaramella, João P. Mendes Tribst, Angelo Aliberti, Pietro Ausiello

**Affiliations:** 1Department of Neurosciences, Reproductive Sciences and Odontostomatological Sciences, University of Naples Federico II, 80131 Naples, Italy; alessandroespedito.dilauro@unina.it (A.E.d.L.); ing.stefano.ciaramella@gmail.com (S.C.); pietausi@unina.it (P.A.); 2Department of Reconstructive Oral Care, Academic Centre for Dentistry Amsterdam (ACTA), Universiteit van Amsterdam and Vrije Universiteit Amsterdam, 1081 LA Amsterdam, The Netherlands; joao.tribst@gmail.com

**Keywords:** finite element analysis, dental materials, resin composite, dental restoration, material properties

## Abstract

This study aimed to investigate the mechanical behavior of resin composites and hybrid glass ionomer cement in class I adhesive dental restorations under loading and shrinkage conditions. Three CAD models of a mandibular first molar with class I cavities were created and restored with different techniques: a bi-layer of Equia Forte HT with Filtek One Bulk Fill Restorative composite (model A), a single layer of adhesive and Filtek One Bulk Fill Restorative (model B), and a single layer of Equia forte HT (model C). Each model was exported to computer-aided engineering software, and 3D finite element models were created. Models A and B exhibited a similar pattern of stress distribution along the enamel–restoration interface, with stress peaks of 12.5 MPa and 14 MPa observed in the enamel tissue. The sound tooth, B, and C models showed a similar trend along the interface between dentine and restoration. A stress peak of about 0.5 MPa was detected in the enamel of both the sound tooth and B models. Model C showed a reduced stress peak of about 1.2 MPa. A significant stress reduction in 4 mm deep class I cavities in lower molars was observed in models where non-shrinking dental filling materials, like the hybrid glass ionomer cement used in model C, were applied. Stress reduction was also achieved in model A, which employed a bi-layer technique with a shrinking polymeric filling material (bulk resin composite). Model C’s performance closely resembled that of a sound tooth.

## 1. Introduction

Bulk resin composite materials, thanks to their improved physical properties, are clinically well-accepted polymeric-based materials for direct posterior restorations to replace missing dental tissues [1]. These materials in combination with resin bonding systems are suggested as idoneous constituents to fully mimic some dental properties, such as aesthetics, resistance, and wear [2].

Notwithstanding this, several points for extensive clinical use of these bulk resin composite polymers in deep and large dental cavities still need to be explored [3]. A dominant factor affecting the mechanical behavior of restored teeth is still the polymerization shrinkage of these resin materials [4]. Reasons that can be highlighted, when considering different dental cavities, are the degree of conversion of the resin monomers, the residual activity of free resin monomers in the organic matrix, the *c-factor*, marginal leakage, and interface debonding [5].

It has been described that another reason for the potential risk of using these dental polymers is the presence of chemical derivates and monomer components that are still biologically active [6,7]. The direct toxic risk mapped by Bis-GMA on human dental pulp cells in vitro, for example, has frequently been described. Despite it being difficult to correlate these research data to clinical behavior, an increase in urinary BPA (Bisphenol A) was reported in patients who received dental treatment with resin-based dental material [8]. This aspect is still being critically investigated [9]. Some research protocols are focused on generating new non-BPA-based monomers in resin composites for dental posterior restorations [10].

Based on that, the wider application of new bioactive dental filling materials as alternatives to posterior resin-based restorations in primary dentition and adult class I teeth has been encouraged [11]. These materials are named alkasite [11] and can release ions, such as F^-^, OH^-,^ and Ca^++^. In this sense, they play a more relevant role at the adhesive interfaces as bioactive compounds than bulk polymeric resin composites, because they can stop the demineralization of dental tissues caused by bacteria of dental caries. On the other hand, they can also clinically be used as bulk materials to fill dental cavities.

However, this class of materials still presents a degree of polymerization shrinkage because of the organic resin base found in its lower traditional light-curing monomers [11].

Glass hybrid restorative materials are part of the new filling dental bioactive class of filling materials, but they are not chemically built as polymers because they react by an acid–base reaction, such as in traditional glass ionomer cement, not by monomer reactions at all. So, they do not shrink at all but show new mechanical properties as dental filling materials [12]. They have been introduced to partially switch the use of bulk resin-based filling dental materials in dentistry to avoid the notorious shrinking effect of all polymeric materials at the tooth–material interfaces, which potentially creates debonding (after stressing) and gap formation with the risk of bacterial leakage and secondary caries. They have been modified by applying only a thin light-cured resin coating agent over the material after setting to better resist the food wear during mastication. Glass hybrid restorative material chemically bonds to enamel and dentine without an extra adhesive layer, is bioactive to remineralize decayed dentine, releases more ions than alkasite materials [12], does not shrink because there are no polymeric monomers inside, and can increase the final resistance of teeth. This study aimed to investigate the mechanical behavior of resin composites and hybrid glass ionomer cement in class I adhesive dental restorations under loading and shrinkage conditions using three-dimensional finite element analysis. The null hypothesis is that there are different behaviors in the use of diverse single- or bi-layer dental materials to replace lost dental tissues.

## 2. Materials and Methods

Various advanced CAD–FEM (Computer-Aided Design and Finite Element Method) techniques have been widely employed to analyze biomechanical responses in dental applications [2,13,14,15]. Starting from these techniques, a 3D CAD model of a sound tooth was designed, and restored models with class I dental cavities (Figure 1) were created.

To account for the variability in masticatory function based on the interaction between the tooth surface and food bolus, the food on the occlusal surface was also modeled (Figure 2). The restored models were designated as A, B, and C and detailed in Table 1. Model A is a bi-layer dental filling material combination based on the use of a polymeric resin composite material, upper layer, and ionic-based hybrid glass ionomer cement lower layer. Model B simulated a posterior dental cavity filled by a single layer (bulk) of polymeric resin composite material, while model C simulated a posterior dental cavity filled by a single layer (bulk) of ionic-based hybrid glass ionomer cement where a thin resin cover was placed. To assess the influence on the mechanical response of Equia Forte HT and Filtek One Bulk Fill Restorative, in a single-layer and a “bi-layer” technique, the models were analyzed using finite element analysis (FEA).

### 2.1. Generation of Solid Models

A healthy lower molar was scanned using a high-resolution micro-CT system (Bruker microCT, Kontich, Belgium) to capture detailed images of the dentin, pulp, and enamel surfaces. The image datasets were then processed with InVesalius software (https://invesalius.github.io/). Three-dimensional tessellated surfaces were created, and cross-section curves were generated thereof. A parametric 3D CAD model was created using loft surfaces in Rhinoceros^®^ (Robert McNeel & Associates, Seattle, WA, USA), and Boolean operations were performed to ensure accurate alignment of the dentin and enamel interfaces. The final tooth CAD model was sectioned 2.5 mm below the cervical area and positioned within a coordinate system, with the X-axis representing the bucco-lingual direction, the Y-axis the mesio-distal direction, and the Z-axis oriented upward (see Figure 1).

A class I cavity, approximately 4.0 mm deep, was created in Rhinoceros^®^. The enamel thickness was about 1.5 mm, and the bucco-lingual and mesio-distal widths were 10.60 mm and 12.36 mm, respectively. The final restored models were developed by applying Boolean operations to the solids.

The food was modeled on the occlusal surface to address the variability in masticatory function caused by the interaction between the tooth surface and food bolus (Figure 2).

### 2.2. Numerical Simulation

Sound and restored models were analyzed by three-dimensional finite element (FE) analysis employing Hyper Works^®^ (https://altair.com/altair-hyperworks, Altair Engineering Inc., Troy, MI, USA) software.

Starting from CAD models, a sound tooth model and three FE models with class I dental cavities were created and investigated using different material combinations, as shown in Figure 1 and summarized in Table 1.

All FE analyses focused on the load during the closing phase of the chewing cycle. Consequently, variability in chewing function was incorporated into the FE analysis modeling of solid food (simulated as apple pulp [16]) on the occlusal surface (Figure 2).

Meshing operations of components of each model were performed in the HyperWorks^®^ environment. Four-node tetrahedral elements (CTETRA) with a global size ranging from 0.05 mm to 0.5 mm were used.

The total number of nodes and elements for the different models of restorations and of sound tooth was 87,993 and 534,084, respectively.

To reduce mesh dependency due to the small radius of curvature and notch effects, mesh refinement techniques were applied. Composite materials used in tooth restorations often exhibit lower shrinkage stresses than those predicted by elastic models. During curing, stress relaxation occurs along with the viscous flow of composites, resulting in a rapid increase in Young’s modulus and viscosity. To account for this, the approach proposed by Kowalczyk [17] was utilized, which adjusts the final shrinkage. For the analyses, an effective linear shrinkage of sr = 0.001 was adopted [17]. Additionally, polymerization shrinkage in adhesive layers and shrinking materials was modeled using the thermal expansion method by applying a one-degree temperature drop.

Nodal displacements at the lower surfaces of the models were fixed in all directions. To simulate physiological masticatory forces [11,18,19], a static occlusal load of 600 N and a transversal load of 20 N were applied vertically and bucco-lingually, respectively. These forces were also analyzed in conjunction with shrinkage effects.

Mechanical properties appointed to each material and the magnitudes of linear shrinkage (%) of shrinking materials are reported in Table 2.

Static linear analyses were carried out considering all materials to behave elastically. So, the analyses were performed considering a non-failure condition.

## 3. Results

The resultant stress distributions for the sound tooth model and three models with class I dental cavities were compared and analyzed.

The considered materials in dental application exhibit brittle behavior, so the maximum normal stress criterion was assumed to assess the potential damage. Thus, the first principal stress was selected to evaluate the results.

Remarkable results were detected in the stress distributions. Figure 3 illustrates the distributions of the first principal stress for enamel, dentin, and restorative material in each model, considering both occlusal and transversal loads alongside the shrinkage effect. Two cross-sections (CS1 and CS2) were analyzed, oriented along the bucco-lingual axis of the tooth. Figure 4 describes global contour plots of the first principal stress for individually investigated models and components.

Quantitative findings were detected by the assessment track, labeled alongside the cavity walls. First principal stresses were mapped by the track and contrasted for the different designs (Figure 5).

As observed in Figure 3, models A and B showed a higher stress concentration than model C. More specifically, models A and B exhibited a high stress concentration at the interface between tooth tissues and restorative material due to polymerization shrinkage. These concentrations were principally located along the enamel–restoration interface for each model and extended in the dentin–restoration interface for model B. For the latter model, stresses arising from loading and polymerization were located internally and marginally (Figure 4).

As seen in Figure 4, models A and B showed the highest stresses in the restoration at the surface near the edge of the cavity, where a value of about 28.5 MPa was reached. The stress peaks propagated from the restoration to the tooth tissue and the highest stresses were located at the interface between tooth tissue and restorative material.

As shown in Figure 5, models A and B displayed similar stress patterns along the enamel–restoration interface, with peak stresses of 12.5 MPa and 14 MPa observed in the enamel, respectively. In model B, stresses also extended to the dentin–restoration interface, where a peak of approximately 9 MPa was recorded. The highest stress gradient within the restoration was found at the top corner of the enamel-dentin interface, where a stress peak of about 22 MPa was detected.

In addition, as observed in Figure 5, models B and C showed a similar trend in stress along the dentine–restoration interface, where a stress peak of 1.25 MPa was found in the enamel for model C, whereas a stress peak of about 0.5 MPa was detected in the enamel of both the sound tooth model and model B.

Therefore, model A, with a shrinking upper layer consisting of a polymeric resin composite and a non-shrinking lower layer consisting of ionic-based glass ionomer cement, showed a similar trend in the stress to model B, only along the enamel–restoration interface, and a very low stress gradient along the dentin–restoration interface and on the cavity floor. So, reduced stress gradients in dentin and the lower restoration layer were evident.

Finally, model C showed a similar sound tooth behavior, with a reduced stress peak of just 1.2 MPa, mainly related to loading and not to the shrinkage effect in dentine and in the restoration itself.

## 4. Discussion

The present analysis was designed to calculate the mechanical behavior of two different filling dental materials employed in a single-layer and a “bi-layer” technique to restore large class I cavities in posterior teeth. The results showed several different behaviors between the combination’s materials, so the null hypothesis was accepted. When a single-increment restoration was simulated, a different mechanical behavior was observed between the Equia Forte HT and Filtek One Bulk Fill in models C and B, respectively, with a numerical difference of 0.92 MPa, not visible in Figure 5 in terms of qualitative stress distribution. Therefore, polymeric- and ionic-based materials employed to restore the same large and deep class I cavity did not show equivalent mechanical behavior when used as a bulk material (single-layer condition) [20]. Figure 5 shows similar mechanical behavior trends along the enamel–restoration interface among different techniques, while a strong difference can be seen at the dentine–restoration interface between models A and B. In the deep dentine area, stress was also different from the sound tooth, and this is in accordance with previous studies where shrinking materials were used in class I and class II adhesive restorations [21,22].

The higher tensile stress observed in Equia Forte HT (model C) suggests a greater susceptibility to failure under load, which could translate to a higher risk of restoration failure or tooth fracture in real-world applications. Conversely, Filtek One Bulk Fill’s performance (model B) indicates potentially more favorable mechanical resilience; however, the observed stress levels still necessitate careful consideration during restoration planning. Future research could benefit from exploring these variations under dynamic loading conditions, simulating masticatory forces to better predict clinical outcomes [23].

Therefore, this study took into account a scenario with optimal adhesion between the restoration and cavity walls, and a non-retentive preparation design was simulated [11]. Despite bond strength being similarly simulated in models A and B at the enamel–restoration material interface, a lower stress distribution was found in deep dentine in model A, probably due to the more flexible and non-shrinking material being placed here (Equia Forte HT). This study confirms that under simulated conditions for large class I adhesive restorations, both material combinations exhibited similar stress patterns on the enamel side. The use of a more flexible base material did not result in better stress relief during the loading.

Low tensile stress was observed in bulk restorations under loading. However, there was only a small difference (about 7.6%) in stress peaks when comparing different restorative materials. Notably, regardless of the base material’s stiffness, the bulk fill resin composite led to lower stress peaks in both enamel and dentin. Given the minimal differences in stress peaks between materials, clinicians should also consider other factors, such as the need for ion release in high-risk patients. Research indicates that alkasite, a bioactive dental filling material with resin monomers, can help prevent caries at the edges of restoration [24,25,26]. In addition, less microleakage is expected when bonding alkasite [27]. In this study, it was shown that perfectly bonded hybrid glass ionomer cement restoration, also a bioactive dental filling material with no resin monomers inside, presents an adaptable stress distribution associated with the adhesion. Previous studies using similar methodologies have utilized maximum principal stress as the criterion for assessing dental material failure [11,16,21]. This approach is based on the failure characteristics of brittle materials, which are prone to crack propagation or interfacial debonding due to high tensile stress concentrations [28]. In this in silico study, the observed stress magnitudes did not reveal significant differences between the restorative materials. However, evaluating their long-term performance remains essential [24]. This study also simulated scenarios with flexible base materials at the cavity’s base, as in model A. Clinicians often include such materials in restorative treatments for large and deep cavities. Previous research has also examined the effects of flowable composites in similar contexts [11]; higher stress levels compared to glass ionomer cement at the enamel tissue were reported. At the dentin margins, stress peaks were similar when using bulk fill resin composite layers. However, models with alkasite, which contains fewer resin monomers, showed smaller stress peaks. Bulk restorations are popular in clinical settings because they are easier to apply and require fewer steps than traditional resin composites [29]. Adding new polymerization modulators and monomers to bulk materials can reduce stress by minimizing volumetric and polymerization shrinkage, as noted in the literature [30,31]. Their translucency and photoinitiators also allow for increments greater than 2 mm [32]. However, in some clinical situations (as simulated in this study), the cavity may already contain flowable composite or glass ionomer cement [33]. Thus, the bulk material will behave in contact with a substrate different from dental tissue. A previous study [11] supported that alkasite can be theoretically proposed since lower shrinking stress peaks were observed in the restoration [11,34]. Conversely, this study showed the mechanical behavior of a restored tooth (model C) with a lower, limited, and more convenient stress distribution than models A and B, focusing on the fact that the hybrid glass ionomer cement investigated (Equia Forte HT) behaved better than other material combinations. Looking at Figure 5, it is possible to see the plotted yellow line for the sound tooth, the green line for model C, and the red line for model B. These two last models’ simulations show the opposite behavior to that of the sound tooth. Model C mechanically behaves closely to the sound tooth model. Further studies are needed to examine the immediate and long-term bond strength between alkasite, hybrid glass ionomer cement, and various other materials. These studies should also verify if their findings align with those of this study.

Considering the potential failure mode, the distinct tensile stress distributions between Equia Forte HT and Filtek One Bulk Fill indicate possible differences in failure susceptibility. The higher tensile stresses observed in these analyses in Equia Forte HT in single-layer restorations (model C) suggest a greater likelihood of material failure or debonding under masticatory forces. This increased stress concentration, especially at the dentine–restoration interface, can initiate crack propagation, leading to premature failure of the restoration in the adhesive interface [35]. In contrast, Filtek One Bulk Fill (model B) which exhibited a more favorable occlusal loading stress distribution, might be less prone to such failures. However, this loading stress concentration in Filtek One Bulk Fill, while lower, could still contribute to the gradual degradation of the material itself over time by releasing unconverted resin monomers (like Bis-phenol A) if not adequately managed through appropriate clinical techniques [36]. The use of the bi-layer technique in this sense could contribute to overcoming these limits, so this technique is preferred. Understanding these failure mechanisms is crucial for clinicians when choosing the appropriate material and restoration technique, as it impacts the longevity and reliability of the restorative work in class I cavities, mainly in young patients.

The findings suggest that, while both materials can be effective under optimal conditions, their long-term performance can diverge due to differences in how they handle stress over time. However, the aging process was not simulated in this quasi-static scenario. Despite that, Filtek One Bulk Fill, with its lower loading stress peaks but unfavorable shrinking stresses, might provide better durability and resistance to the stresses of daily mastication and mechanical cycling. This could lead to potentially longer-lasting restorations. Equia Forte HT, on the other hand, may offer advantages such as ion release and caries inhibition, which can be particularly beneficial in high-risk patients, but its higher stress levels suggest a need for careful monitoring and possibly more frequent maintenance or replacement [37].

This study, constrained by the limitations of 3D finite element analysis, did not account for all factors present in the oral environment [11,21,38]. Factors such as pH and temperature fluctuations, potential defects in the adhesive layer or restorative material, and varying chewing loads were not considered. Additionally, the analysis assumed ideal adhesive bonding strengths for all materials, without incorporating additional model validations through in vivo or in vitro testing, which could offer more clinically relevant insights [39,40]. Moreover, the food bolus was modeled as a single volumetric body, which does not fully capture the diversity of a patient’s diet and the range of loading conditions.

## 5. Conclusions

This analysis highlights the importance of minimizing shrinkage effects in dental fillings to achieve biomimetic behavior in posterior restorations. The in silico study revealed that non-shrinking materials, such as the hybrid glass ionomer cement, significantly reduced stress in deep class I cavities in molars. The bi-layer technique with bulk resin composite also showed stress reduction.

## Figures and Tables

**Figure 1 polymers-16-02525-f001:**
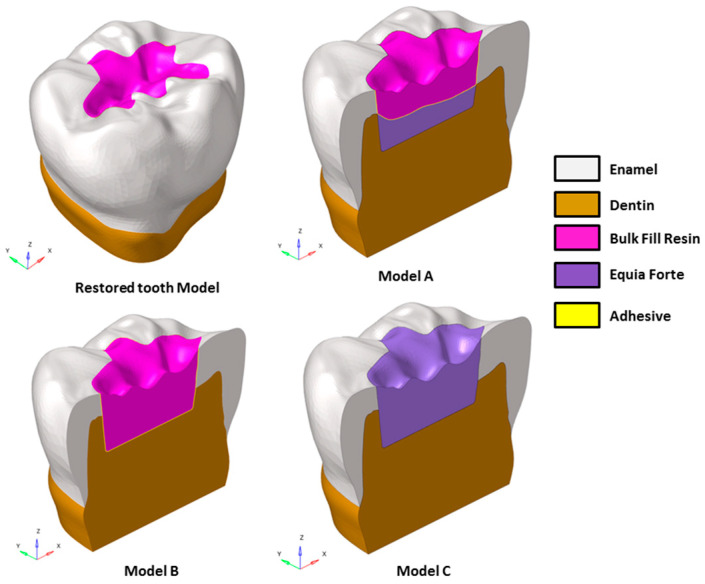
Three-dimensional CAD model of a restored tooth, incorporating three variations of class II mesio-occlusal cavities.

**Figure 2 polymers-16-02525-f002:**
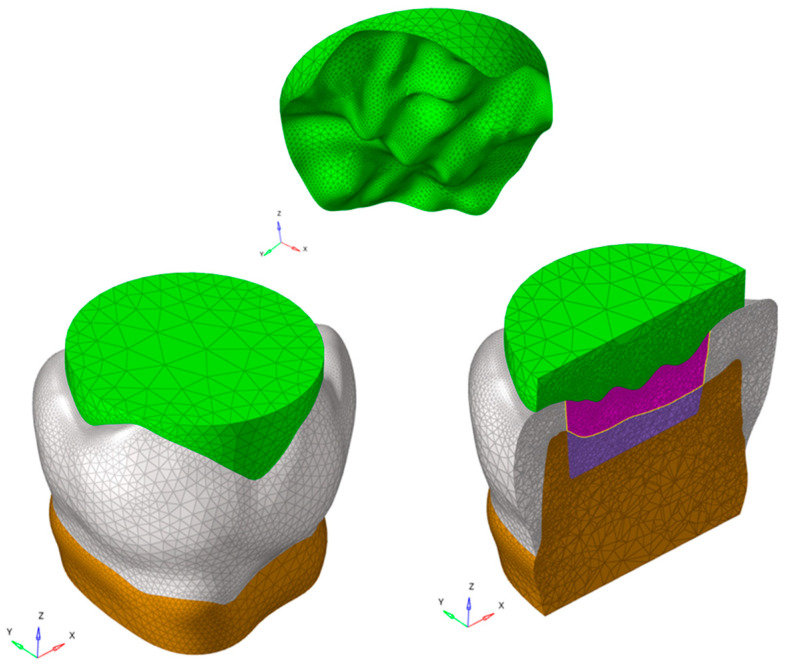
Solid food was modeled on the occlusal surface to simulate the contact between the tooth surface and the food bolus.

**Figure 3 polymers-16-02525-f003:**
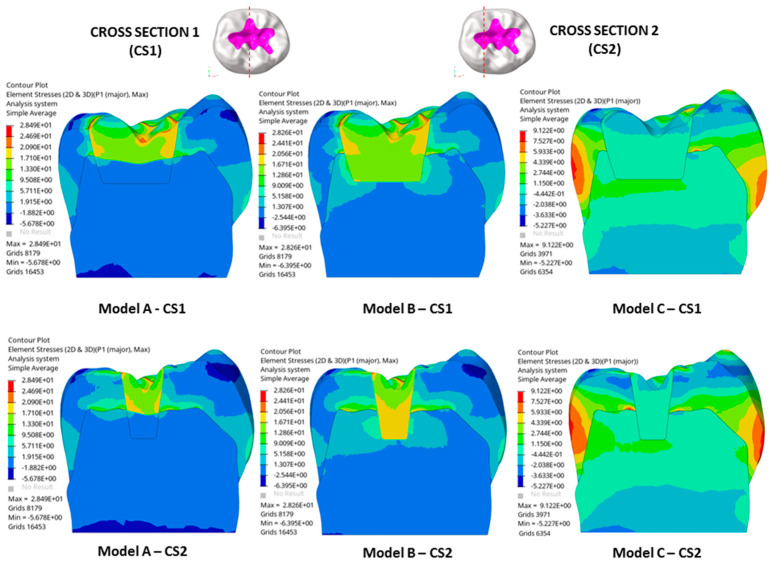
Stress distributions for the first principal stress in enamel, dentin, and restorative material are shown for each model, considering both occlusal and transversal loads as well as the shrinkage effect. The analysis includes two cross-sections taken along the bucco-lingual axis of the tooth.

**Figure 4 polymers-16-02525-f004:**
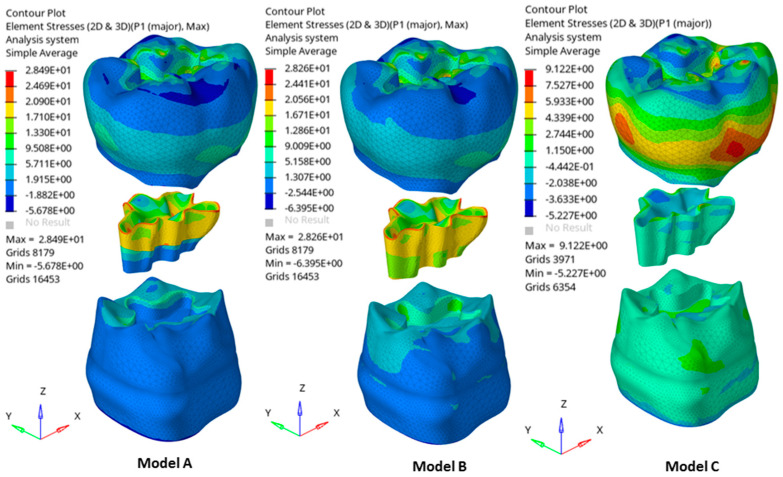
Global contour plots of the first principal stress for each design.

**Figure 5 polymers-16-02525-f005:**
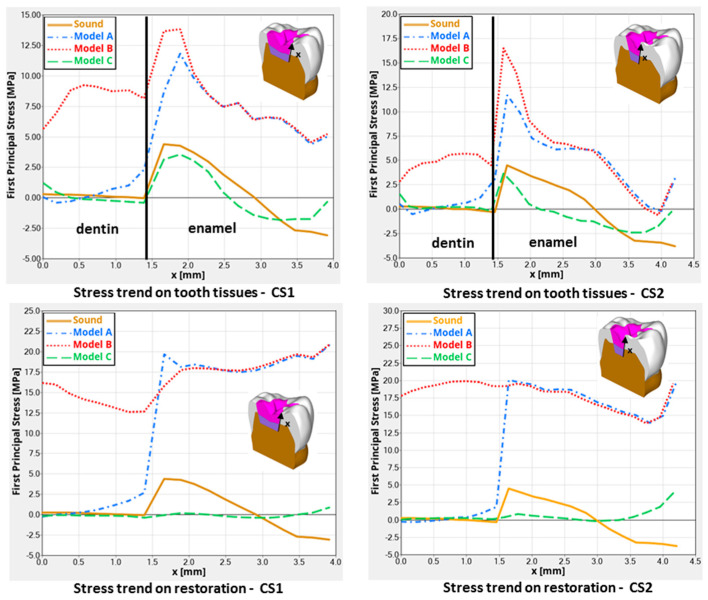
The first principal stresses were charted along the assessment track and compared among the various models. Each model displayed a high stress magnitude at the interfaces between tooth tissues and the restorative material.

**Table 1 polymers-16-02525-t001:** Restorative material combinations and thickness.

Model	Adhesive Layer	Restorative Lower Layer	Restorative Upper Layer	Single Restorative Material
A	10 µm thick (only around restorative upper layer)	Equia Forte HT (about 1.45 mm thick and 20 mm^3^ volume)	Bulk fill resin composite (about 2.5 mm thick and 33 mm^3^ volume)	-
B	10 µm thick	-	-	Bulk fill resin composite (about 4 mm and 53 mm^3^ volume)
C	-	-	-	Equia Forte HT (about 4 mm and 53 mm^3^ volume)

**Table 2 polymers-16-02525-t002:** Mechanical properties of materials: Young’s modulus, Poisson’s ratio, and linear shrinkage. Data reported by [11,17].

Material	Young’s Modulus (GPa)	Poisson’s Ratio	Linear Shrinkage Ratio (%)
Dentin	18.00	0.23	--
Enamel	80.00	0.30	--
Food (apple pulp)	3.41	0.1	--
Adhesive layer	4.00	0.30	1.0
Bulk fill composite	12.00	0.25	1.0
Equia Forte HT	6.27	0.25	--

## Data Availability

The original contributions presented in the study are included in the article, further inquiries can be directed to the corresponding author.

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
