# Peer review of "Comparison of Bulk Polymeric Resin Composite and Hybrid Glass Ionomer Cement in Adhesive Class I Dental Restorations: A 3D Finite Element Analysis"

_polymers, 2024, doi:10.3390/polym16172525_

Round 1
Reviewer 1 Report
Comments and Suggestions for Authors
The article "Polymeric resin composite or hybrid glass ionomer cement in adhesive class I dental restorations? A 3D-FEA" by A.E. Di Lauro and colleagues is devoted to the study of tooth stress distribution under the external mechanical load after dental restoration. The study was performed using calculation methods (FEM), and the article considered several variants of 3D-models, suggesting various combinations of dental restorative materials. As a result of the research, the conclusion was reached that in order to minimize the stresses concentration, it is necessary to use materials with minimal shrinkage.
There are several questions and comments on the paper.
1. The main conclusion of the article, which is written in the "Conclusion" section, is about the importance of using materials with low shrinkage due to reduction the stresses concentration in the contact zone. At the same time, the article used only one material with low shrinkage (model C), which, moreover, differs in elastic properties from the materials used in other models (A and B). In this regard, making such a general conclusion based on a three combinations is probably not very justified.
In addition, the point regarding the magnitude of the shrinkage is unclear. Where the data on shrinkage in Table 3 came from? Also, in lines 118-121 it is said that a certain "simplified approach" was used to estimate the shrinkage, on the basis of which it was chosen equal to 0.001. If the shrinkage is such an important parameter that fundamentally affects the distribution of stresses in the contact zone, then how can we use some "simplified approach" to estimate it? It is also not entirely clear from the paper text how the data in lines 118-121 were used when setting the properties of materials in the model.
2. Table 2 is uninformative, it should be removed from the article, and the information presented in it should be changed by one line of text.
3. The share of self-citations in the references list is quite large - about 20%. Please, reduce in to the value of < 15%.
4. The model considers the materials as absolutely elastic bodies, without plastic deformation. But do the real materials studied in the research have a plasticity margin? And if we take into account the possibility of local plastic deformations, this can greatly change the stress distribution.
5. In line 220, the materials “BF” and “Alk” appear. What are these materials and why are they mentioned in the article? These abbreviations were not used in the paper earlier. Please, clarify this point.
6. In the absence of verification of the developed model, it is difficult to judge the reliability of the results obtained. It would be desirable authors to provide comments on this point in some way.
___
I believe that the article should be reviewed again after the authors make the appropriate changes to the article and provide comments.
Author Response
Thank you very much for taking the time to review our manuscript. Please find the detailed responses below and the corresponding revisions and corrections highlighted in the re-submitted file
Reviewer n. 1
The article "Polymeric resin composite or hybrid glass ionomer cement in adhesive class I dental restorations? A 3D-FEA" by A.E. Di Lauro and colleagues is devoted to the study of tooth stress distribution under the external mechanical load after dental restoration. The study was performed using calculation methods (FEM), and the article considered several variants of 3D-models, suggesting various combinations of dental restorative materials. As a result of the research, the conclusion was reached that in order to minimize the stresses concentration, it is necessary to use materials with minimal shrinkage.
There are several questions and comments on the paper.
- The main conclusion of the article, which is written in the "Conclusion" section, is about the importance of using materials with low shrinkage due to reduction the stresses concentration in the contact zone. At the same time, the article used only one material with low shrinkage (model C), which, moreover, differs in elastic properties from the materials used in other models (A and B). In this regard, making such a general conclusion based on three combinations is probably not very justified.
Response:
We thank you to point out these aspects. We have modified as suggested the Conclusion section where we rewrote better the sense of the results in according to your focus. The Conclusion is that the most convenient mechanical behavior of the three different investigated model of dental posterior restorations (differently restored by polymeric and no polymeric material combinations) in term of stresses arising from shrinkage and loading is related to model C. Here, anyway, the simulation by means of 3D-FEA has considered the use of bulk (single layer) no shrinking material (Equia Forte HT, as reported in Tab. 2, is a polymer-free dental filling material).
1 bis. In addition, the point regarding the magnitude of the shrinkage is unclear. Where the data on shrinkage in Table 3 came from? Also, in lines 118-121 it is said that a certain "simplified approach" was used to estimate the shrinkage, on the basis of which it was chosen equal to 0.001. If the shrinkage is such an important parameter that fundamentally affects the distribution of stresses in the contact zone, then how can we use some "simplified approach" to estimate it? It is also not entirely clear from the paper text how the data in lines 118-121 were used when setting the properties of materials in the model.
Response:
We agree with your comment. It is a known that measured residual stresses in cured dental resin composite materials are much smaller than those calculated on the basis of the elastic model. This fact is due to the changes of the elastic properties and the stress relaxation accompanying the viscous flow of resin composites, during the curing process. To take into account the above fact, as demonstrated by Piotr Kowalczyk in the citated paper [18], the final shrinkage smax should be reduced according to the rule: sr=(σr/σ)smax. In accordance with what is reported in Piotr Kowalczyk's paper, the effective residual stress σr can be estimated about 10% of the elastic stress σ, calculated considering the linear shrinkage smax (1.0%). So, the value of effective linear shrinkage, adopted in this work, is sr = 10% smax = 0.001. Polymerization shrinkage for the adhesive layers and shrinking materials was simulated with the thermal expansion approach by setting a one-degree drop in temperature. The linear thermal expansion coefficient for shrinking materials was assumed equal to 0.001.
- Table 2 is uninformative, it should be removed from the article, and the information presented in it should be changed by one line of text.
Response:
We thank you; your comment is right, and we have removed Table 2 while the informations were typed in the text at lines 122-123.
- The share of self-citations in the references list is quite large - about 20%. Please, reduce into the value of < 15%.
Response:
We thank you for this comment. Done. Now the share of self-citations was reduced to < 15%, as suggested, where n. 12, 13, 15, 19, 25 were replaced (in red) by other convenient references in the final list.
- The model considers the materials as absolutely elastic bodies, without plastic deformation. But do the real materials studied in the research have a plasticity margin? And if we take into account the possibility of local plastic deformations, this can greatly change the stress distribution.
Response:
Thanks for the comment. Materials in dental application exhibit brittle behavior and are characterized by their tendency to fracture with little or no plastic deformation when subjected to stress. Unlike ductile materials, which can undergo significant deformation before failure, brittle materials fail suddenly and catastrophically. So, the elastic approach in dental application allows for detailed analysis of stress distribution, deformation, and potential failure points in restored teeth under functional loads.
- In line 220, the materials “BF” and “Alk” appear. What are these materials and why are they mentioned in the article? These abbreviations were not used in the paper earlier. Please, clarify this point.
Response:
We thank you again for the comment. It was wrongly mentioned the class of materials. We appreciate it and we have cleaned the abbreviations at lines 230, 241, 242, 253, 256, 266, 270, of the text; we also have better explained the meaning of the different materials combination.
- In the absence of verification of the developed model, it is difficult to judge the reliability of the results obtained. It would be desirable authors to provide comments on this point in some way.
Response:
We thank you to focus this point. It is right, the model validation is an important complementary aspect of all FEM analyses, also if it is not always easy and right to obtain [40], for example in some biomedical applications. The results obtained in this paper were compared with the same reported in similar works. For instance, the stresses calculated in the present work are of the same magnitude of the values reported in Piotr Kowalczyk's paper. So, the results of the analyses are considered reliable. In the text we have better reconsidered this aspect, lines 313-314. No sentences were carried out.

Reviewer 2 Report
Comments and Suggestions for Authors
The manuscript undertook the Polymeric resin composite or hybrid glass ionomer cement in adhesive class I dental restorations? A 3D-FEA. This study focus on the mechanical behavior of resin composites and hybrid glass ionomer cement in class I adhesive dental restorations. The manuscript lacks of novelty.
1. I did not find any novelty of the present study, I suggest please incorporate in the introduction section.
2. Introduction should be rewrite, its difficult to understand why authors choose resin composites and hybrid glass ionomer cement. What is the existing issues and how authors overcome this issue.
3. Authors should mention the details of models used in this study.
4. What model B and C suggested??????????
5. Establish the relationship between A, B, and C model....
6. Conclusion should be rewrite.
Author Response
Thank you very much for taking the time to review our manuscript. Please find the detailed responses below and the corresponding revisions and corrections highlighted in the re-submitted file.
- I did not find any novelty of the present study; I suggest please incorporate in the introduction section.
Response:
Thank you for your comment. The introduction section has been revisited as suggested and has been clarified the news in lines 49-54, on one side, and in lines 58-68 on the other side.
- Introduction should be rewrite, its difficult to understand why authors choose resin composites and hybrid glass ionomer cement. What is the existing issues and how authors overcome this issue.
Response:
Thank you for the comment. We have highlighted the points now, revisiting in the Introduction section , lines 51-55, the concept that in Dentistry it is important to rebuild the lost function of decayed teeth by dental materials which can mechanically behave as close as possible to dental tissues. To overcome the limits of some materials it is important to check new filling materials and different therapeutical strategies, lines 59-71.
- Authors should mention the details of models used in this study.
Response:
Thanks for the right comment. We agree with you, and it was done in the Materials and Methods section, filling the gap in lines 87-92.
- What model B and C suggested??????????
Response:
We thank you for the comment and we tried to point out now it better in Results (lines 192-194-198) and Discussion (238-251; 256-265;) what you asked for. Mechanical behavior of restored teeth in simulated B and C models suggested that the choice of different dental materials can deeply influence the final stress distribution (shrinkage and loading) than the sound tooth behavior. Looking at the Fig. 5, it is possible to see the plotted line yellow for sound tooth and the green for C model while the red for B model. These two last model’s simulations are the opposite behavior than the sound tooth. C model behaves better than B model
- Establish the relationship between A, B, and C model....
Response:
We agree with your point. The relation among A, B and C models was focused now on lines 275-280. In these analyses by means of FEA, it was investigated the mechanical behavior of traditional (polymeric-based) and new (ionic-based) dental filling materials and their combination to restore teeth after enamel and dentine lost in large and deep class I cavities (in posterior teeth). The use of materials with different elastic modulus (E) and their combination has been investigated under stressing conditions, occlusal loading and polymerization shrinkage, by means of finite element analysis. Within the postulated limits of the methodology itself, the results indicated model C mechanically behaves better than model A than model B.
- Conclusion should be rewrite.
Response:
Thanks for the comment and help. We did it, lines 319-324, according to your suggestions.

Round 2
Reviewer 1 Report
Comments and Suggestions for Authors
The article "Polymeric resin composite or hybrid glass ionomer cement in adhesive class I dental restorations? A 3D-FEA" by A.E. Di Lauro and colleagues (revised version).
All main critical points of the paper were corrected by the authors in the new revised version. The authors provided detailed explanations to the comments and made appropriate changes to the article. The share of self-citations was reduced to the required values. Probably, the conclusion section may be expanded, but this is not a critical point.
No further observations from my side. I consider that the article can be published in "Polymers"
Author Response
Thank you!
Reviewer 2 Report
Comments and Suggestions for Authors
Accept
Author Response
Thank you!